# Suicidal Ideation in the Australian Construction Industry: Prevalence and the Associations of Psychosocial Job Adversity and Adherence to Traditional Masculine Norms

**DOI:** 10.3390/ijerph192315760

**Published:** 2022-11-26

**Authors:** Simon Tyler, Kate Gunn, Adrian Esterman, Bob Clifford, Nicholas Procter

**Affiliations:** 1Mental Health and Suicide Prevention Research and Education Group, UniSA Clinical and Health Sciences, University of South Australia, Adelaide, SA 5001, Australia; 2Department of Rural Health, University of South Australia, Adelaide, SA 5001, Australia; 3UniSA Allied Health & Human Performance, University of South Australia, Adelaide, SA 5001, Australia; 4MATES in Construction South Australia, Adelaide, SA 5034, Australia

**Keywords:** suicide, suicidal ideation, construction industry, masculinity, psychosocial job adversity

## Abstract

Background: Suicide in the Australian Construction Industry (ACI) is a significant issue, however minimal understanding of suicidal ideation prevalence, as well as the potential role psychosocial job adversity and increased adherence to traditional masculine norms may play in its presence, is apparent. Method: A representative sample of Australian men (*n* = 11,132) were used to create initial understandings of prevalence of suicidal ideation (past two weeks), psychosocial job adversities and level of adherence to traditional masculine norms for the ACI (*n* = 1721) in comparison to a general population comprised of the remaining employed males from Other Industries (*n* = 9411). Additionally, due to their reported increased suicide vulnerability investigation of associations between suicidal ideation, psychosocial job adversities and adherence to traditional masculine norms for the ACI were undertaken. Results: No difference in suicidal ideation prevalence was reported between the ACI and those employed in Other Industries (*p* > 0.05), however, increased prevalence of psychosocial job adversities (*p* ≤ 0.001) and adherence to traditional masculine norms (*p* ≤ 0.001) for the ACI was seen. Significant multivariate associations between suicidal ideation, psychosocial job adversities (OR = 1.79, 95%CI [1.12–2.85]) and two domains of traditional masculine norms, self-reliance (OR = 1.29, 95%CI [1.09–1.51]) and risk-taking (OR = 1.20, 95%CI [1.01–1.41]), were reported. Conclusion: Results suggest need for increased understanding of later stage suicidal trajectory drivers in the ACI. Findings indicate need for prevention group/industry concentration on mitigation of psychosocial job adversities, as well as a more nuanced and increased discussion of the negative role of self-reliance and risk-taking domains of traditional masculine norms may play in ACI suicidal ideation, as opposed to the construct as a whole.

## 1. Introduction

A total of 3144 Australians died by suicide in 2021 and each year approximately 400,000 are exposed to or impacted by suicidal behaviour [1]. With suicide outcomes resulting in significant negative consequences for those bereaved as well as for the economy, the need for information that can assist in the development and delivery of targeted suicide prevention programs is vital.

A range of population level drivers of suicide, including but not limited to, socioeconomic status, previous trauma, mental health condition presence and employment in certain occupations/industries, have been highlighted and assist in identifying those of greater need of intervention [2]. Gender differences are also apparent, with men accounting for seven of the nine suicides that occur every day in Australia [1]. With men over-represented among those who die by suicide, it is unsurprising that those employed in the male dominated Australian Construction Industry (ACI) are regularly reported to have increased suicide vulnerability, in comparison to many other populations [3,4,5]. However, despite the role male dominance may play in the increased suicide vulnerability regularly reported for the ACI, it is also suggested that there are a range of other factors unique to the ACI that play a role in this disparity and as a result the ACI are a population that requires specific focus to understand what may drive these outcomes so as to inform preventative programs and industry [3,4,5]. Programs aimed at reducing this disparity have been developed and implemented, with a level of success reported [6].

Despite these intervention efforts, the issue remains significant and is further complicated by a distinct lack of literature and understanding regarding drivers of suicidal trajectories for this population. While there is largely widespread agreement regarding the limited predictive validity of suicide from a structured risk assessment, there is acceptance of early stages of the suicidal trajectory, namely suicidal ideation, as a useful identifier of future suicide risk [7,8,9,10,11]. As a result, scholars suggest knowledge and mitigation of the drivers of suicidal ideation, independent from suicidal behaviours and attempts, is a key component in effective suicide prevention [8,12]. Given the significant number of suicide outcomes reported in the ACI, knowledge of suicidal ideation prevalence and its potential drivers, may inform and guide suicide prevention efforts aimed at mitigating trajectories towards suicide for this population.

Previous research has proposed a wide range of potential drivers of suicide in the global construction industry, including commonly understood bio-psychosocial factors such as socioeconomic status, relationship status, drug and alcohol misuse and age [13,14,15,16]. However, while these factors may play a role in the presence of suicidal ideation for the ACI, as shown in other populations, less than desirable empirical evidence is available and a lack of research focused on potential industry drivers is also evident [13,14]. In fact, while recent systematic and literature reviews of the area highlight the potential role of a number of industry factors, little to no research has moved past theoretical suggestion of the role may play [13,14]. Furthermore, recent reviews highlight the need for research to account for potential cultural and socio-political differences within which the industry functions, suggesting national research streams are imperative to create more rigorous understandings [13,15].

Earlier research in the area has theorised psychosocial job adversity, such as presence of decreased level of job security or job control, may play a role in the presence of suicidal ideation for ACI workers [3,5]. However, despite research in other populations suggesting relevance, to the best of the authors’ knowledge no discrete formalised investigation has taken place focussing on the prevalence of psychosocial job adversities experienced by the ACI, nor its association with suicidal ideation [14,15].

Additionally, adherence to traditional masculine norms, is also regularly suggested as playing a key role in increasing risk of suicide and suicidal ideation for those in the ACI [5,13]. The conceptual framework that informs this suggestion is that many men engage with these certain roles, attitudes, behaviors, and meanings considered appropriate by a society or culture’s ‘dominant’ and ‘traditional’ representation of what it is to be, or is expected of, a man [17]. Due to social conventions and the desire to “fit in”, it has been suggested that men will adhere to these norms with subsequent belief that due to the high levels of male employment in the ACI, a culture may have developed that further encourages adherence to traditional masculine norms [5,13,17,18]. Broader research has proposed the role of adherence to traditional masculine norms in male suicidal behaviours, suggesting it may drive gender differences seen in suicidality, including increased use of lethal means by men, potential for restricted emotionality regarding serious (e.g., presence of mental health conditions or relationship breakdowns) and increased levels of stoicism that may impact help seeking behaviors [5,19,20,21]. Despite this, little is known about whether this proposed culture of increased adherence to traditional masculine norms exists within the ACI. Furthermore, there is contention around whether adherence to traditional masculine norms is exclusively associated with negative outcomes and researchers have rightly highlighted, the reductionist stance in which traditional masculine norms contextualizes masculinity [20,21,22]. Less than desirable research has delved deeper and investigated whether there are only certain domains associated with negative outcomes such as suicidal ideation presence. For example, while previous research in Australian male populations has shown adherence to the traditional masculine norm domain of self-reliance is associated with suicidal ideation presence, other domains, such as primacy of work, do not show associations [19,23,24]. Therefore, while masculinity is commonly suggested as a significant driver of suicidal behaviors for the ACI based on its cultural endorsement of adherence to traditional masculine norms, empirical evidence is needed to support this claim as well as create a more nuanced understanding. To the best of the authors’ knowledge no research has attempted to understand the potential differences in level adherence to traditional masculine norms for those employed in the ACI, nor the associations of between the specific domains and the presence of suicidal ideation.

As a result, the current study, investigates the prevalence of suicidal ideation and psychosocial job adversities in a representative sample of Australian men working in the ACI, as well as levels of adherence to traditional masculine norms, through comparison to a general population of men employed in other industries create initial understandings of whether there are significant differences for the ACI regarding these issues. Based on previous research suggesting increased suicide vulnerability for the ACI, as well as the potential role of psychosocial job adversities and increase adherence to traditional masculine norms in these outcomes, it is hypothesised increased prevalence of suicidal ideation and psychosocial job adversities, as well as greater endorsement of traditional masculine norms, will be seen in the ACI, in comparison to males employed in Other Industries [3,4,5,13]. Additionally, this study explores a range of potential drivers of suicidal ideation presence in the ACI, with a focus on two regularly implicated industry factors, psychosocial job adversity and adherence to traditional masculine norms, to understand their relevance and inform prevention programs aimed at mitigating suicidal trajectories in this vulnerable population. Based on previous research, it is hypothesised that psychosocial job adversity will be associated with suicidal ideation in ACI employees, as will adherence to traditional masculine norms, however, the domain of self-reliance will explain the majority of this association [19,23,24].

## 2. Materials and Methods

### 2.1. Data Source

To assess the prevalence of suicidal ideation and associations with masculinity and psychosocial job factors, the current research undertaken in October 2022 utilised cross-sectional data from the baseline wave (Wave 1, 2013–14) of *Ten to Men*. *Ten to Men* is a representative sample of men from all Australian states and territories, inclusive of both regional and city populations with the longitudinal study designed to help improve the health and well-being of boys and men [19,25]. The procedures and materials used in the *Ten to Men* data collection have been described in more detail elsewhere, with additional information visible on the study’s website (http://www.tentomen.org.au/, accessed on 1 October 2022). Briefly, of the 15,988 participants who returned useable data in the baseline wave (Wave 1) of *Ten to Men*, we initially included 13,896 males aged between 18–55 years old. This sample was further reduced excluding those who reported unemployment (remaining *n* = 11,607) and then those who did not provide complete data on key variables of interest (remaining *n* = 11,132). Participants were recruited in 2013–2014 using a stratified, multi-stage, cluster random sampling strategy in which the primary unit of sampling was the household. Initially, 104,884 households in 622 randomly selected statistical areas, ensuring inclusion of those with broad backgrounds and life experiences, were identified for potential participation. Contact with 81,400 (78%) of these households was achieved and 33,724 (32%) were identified as having at least one in-scope resident (a male aged between 10 and 55). Identification of 45,510 in scope males in these households was made, providing representation from all Australian states and territories and from major cities and inner and outer regional areas. Participants provided information via self-report questionnaires on domains including physical health, mental health and wellbeing, health behaviours, social determinants of health, and health service utilisation and health knowledge. It is important to note that there have been future waves of data from the *Ten to Men* cohort released; however, adherence to traditional masculine norms was not assessed beyond Wave 1. While longitudinal analysis was considered there are potential methodological flaws in treating adherence to traditional masculine norms as a stable construct. For this reason, cross sectional investigation of the prevalence of suicidal ideation, psychosocial job adversities, level of adherence to traditional masculine norms, as well as the associations between psychosocial job quality, masculinity, and suicidal ideation, while controlling for a number of confounding variables, was the chosen methodological approach. Figure 1 provides a visual representation of the methodology process undertaken.

### 2.2. Measures

#### 2.2.1. Suicidal Ideation

The key outcome variable for this study, suicidal ideation, was derived from participant responses to single item on the Patient Health Questionnaire included in the survey [26]. This item asked participants ‘‘Over the past two weeks, how often have you been bothered by thoughts that you would be better off dead, or of hurting yourself in some way?’’ Responses are scored 0 (‘‘not at all’’), 1 (‘‘several days’’), 2 (‘‘more than half the days’’) or 3 (‘‘nearly every day’’). Similar to other studies using the same database for the current analysis a dichotomous variable was created with those who reported scores of 1 or more classified as currently experiencing suicidal ideation (coded as 1 and those without suicidal ideation presence coded as 0) [19,23].

#### 2.2.2. Employment Classification

Participants were classified as Australian Construction Industry (ACI) employees based on their response to the question “What is the main business, industry, or service of your employer? If you are self-employed, what is the main business, industry, or service of your business?” One of the survey response options was “Construction” and those who responded this way were classified as ACI employees (coded as 1). Those who provided alternative responses were classified as being employed in Other Industries (coded as 0), so long as they did not indicate unemployment. Other Industries comprised of those employed in accommodation and food services, administrative and support services, agriculture, forestry and fishing, arts and recreation services, education and training, electricity/gas/water/waste services, financial and insurance services, health care and social assistance, information media and telecommunications, manufacturing, mining, personal and other services, professional/scientific/technical services, public administration and safety, rental/hiring/real estate services, retail trade, transport/postal/warehousing and wholesale trade. Those reporting unemployment at an earlier question in the survey were excluded from the study due to research demonstrating a significant disparity in suicide risk and rates for those unemployed, particularly long term unemployed, which would likely have resulted in distortion of results [27].

#### 2.2.3. Psychosocial Job Adversity

Responses to questions, with response options ranging from “strongly disagree” (1) to “strongly agree” (6), regarding participant perceptions of their level of job control, job demands/complexity, job insecurity, and unfair pay were used to compute a dichotomous measure of presence of psychosocial job adversity (full list of questions posed included in Appendix A). In short, participant responses to questions relating to each of the above listed factors of overall psychosocial job quality, were summed, with dichotomized variables created based on the upper quartile experiencing the greatest adversity. A new dichotomized variable was then created with those experiencing psychosocial job adversity in any domains of job control, job demands/complexity, job insecurity, or unfair pay classified as experiencing a psychosocial job adversity (coded as 1) and those not experiencing a psychosocial job adversity (coded as 0). The construction of the psychosocial job quality variable was based on development of a similar variable of psychosocial job quality used in previous research with the *Ten to Men* cohort [28,29,30].

#### 2.2.4. Adherence to Traditional Masculine Norms

Adherence to traditional masculine norms was assessed via participants’ responses to the highly popularised Conforming to Masculine Norms Inventory (CMNI-22) designed to measures cognitive, behavioural, and affective conformity to traditional/hegemonic masculine norms [23]. The CMNI-22 is an abbreviated version of the original 94-item scale and uses the two highest loading statements from each specific masculine norm domain to assess subscale endorsement [23,31]. These pairs of statements correspond to 11 subscales: (1) Primacy of Work; (2) Dominance; (3) Risk-Taking; (4) Heterosexual presentation; (5) Power over Women; (6) Emotional Control; (7) Playboy; (8) Violence; (9) Pursuit of Status; (10) Winning; and (11) Self-Reliance. The CMNI instructs respondents to consider their actions, feelings, and beliefs when rating their agreement or disagreement with each of the 24 statements. Response options range from “strongly disagree” (0) to “strongly agree” (3). Responses to each item are summed to provide continuous conformity scores for each subscale ranging from 0 to 6, as well as summation of total subscale scores to present a continuous score of overall conformity to masculine norms ranging from 0 to 66 (higher scores indicating greater adherence to masculine norms) [31]. When the original 94-item instrument was developed it was shown to have good internal consistency, construct validity and discriminant validity [19]. The 22-item instrument demonstrates excellent concurrent validity, correlating well with the original instrument [19,23,31]. Full list of questions posed by the CMNI 22 located in Appendix A.

#### 2.2.5. Other Variables

Other variables that could potentially confound the relationship between masculinity and suicide ideation were age, marital status, significant life events, socio-economic status, alcohol use, country of birth and various current mental health states. These covariates were considered based on previous analysis of associations between suicidal ideation presence, masculinity, and psychosocial job quality in men, as well as the fact they can be considered as accounting for the majority of suggested drivers of suicidal ideation in the ACI highlighted in a previous systematic review of the area [15,19,23,29]. Age was treated as a continuous variable [4]. Responses to a single question asking participants’ country of birth were transformed into a dichotomous variable (those reporting being born in Australia coded as 0—born in country other than Australia coded as 1) [23]. Socio-economic status was transformed into an ordinal variable (1 greatest disadvantage–5 least disadvantage) with participants ascribed to levels on the basis of their area of residence, using quintiles from the Index of Relative Socioeconomic Disadvantage [19]. Relationship status was transformed into a dichotomized variable (married/de facto coded as 0; never married/widowed/divorced/separated coded as 1) [4]. A dichotomised variable was created for alcohol consumption based the Alcohol Use Disorders Identification Test (AUDIT) classification of alcohol use as harmful/hazardous or not (appropriate alcohol consumption coded as 0—inappropriate coded as 10 [32]. The various current mental health statuses were based on self-reported presence of depression, post-traumatic stress disorder, anxiety and schizophrenia, determined by responses to a question that was part of a broader set of assessment of presence of health conditions based on the Australian Health Survey [19]. Dichotomised variables for each mental health conditions were constructed based on participant responses to the question ‘‘Have you been treated for or had any symptoms of (depression, PTSD, anxiety disorder, schizophrenia) in the past 12 months?’’ with ‘‘yes’’ or ‘‘no’’ response options (no condition presence coded as 0—condition presence coded as 1). Exposure to a stressful life event in the last twelve months was assessed via responses to six life events shown previously as relevant to suicidality being serious personal injury, illness, or surgery; break-up of a serious relationship/divorce/separation; serious conflict with a family member; difficulty finding a job; legal troubles or involvement in a court case; and major loss or damage to personal property [33,34,35]. Participants indicated whether any of these events had taken place in their lives over the past 12 months and responses were transformed into a dichotomised variable (no stressful life events coded as 0—presence of any of the six stressful life events coded as 1).

### 2.3. Statistical Considerations

Those reporting unemployment, along with those who did not respond to the three key variables of interest (suicidal ideation, masculinity, and psychosocial job adversity), were removed from analysis. To assess the prevalence of suicidal ideation for those employed in the ACI, a chi square test of homogeneity was utilised to make comparisons between those employed in the ACI and those employed in Other Industries. The use of those employed in Other Industries as a comparator was to allow for comparison to a general population exposed to alternative work environments, however not compromised by the well-known risks that are associated with unemployment. The same statistical process was undertaken to assess prevalence differences of psychosocial job adversity for those employed in the ACI and those employed in Other Industries A Mann–Whitney U test was run to determine if there were differences in endorsement of traditional masculine norms (total score across domains as measured by the CMNI) between those employed in the ACI and those employed in Other Industries. Following this, data were restricted to those classified as employed in the ACI. Univariate logistic regression analyses to assess the strength of association between suicidal ideation presence and the key variables of interest, being traditional masculine norm adherence and psychosocial job adversity, were undertaken. For multivariate analysis, all variables of a continuous nature were assessed for linear relationships with the outcome variable (transformations applied if necessary and used in subsequent analysis) and all variables were assessed for collinearity. Following this, all variables, including those of primary interest, were subjected to bagging, a process of stepwise bootstrap aggregation, to reduce the list of variables. The aim of bagging is to reduce the inclusion of irrelevant variables and improve the overall model accuracy by using a combination of bootstrapping and stepwise logistic regression modeling to generate multiple model versions, then summarize the number of times variables were used within each model [36]. This process involves a number of stages including removal of variables with high levels of missing data as these may indicate a data collection issue, as well as potentially impact model stability and strength [37,38]. Following removal of variables with high levels of missing data, One thousand bagging episodes were undertaken and variables that were included less than 300 times were excluded in the initial round. This bagging process was repeated a second time and again any variables used less than 300 in the one thousand bagging episodes were excluded. Following this logistic regression with remaining variables was undertaken to obtain coefficients and assess if any were reported as insignificant (this did not occur so further analysis was no required). The final logistic regression model was assessed for outliers (studentized r value > than 3 suggestive of presence). A test of model specification using the Stata Linktest procedure was undertaken [39]. Finally, a Hosmer-Lemeshow goodness-of-fit test was conducted [40]. Initial data coding, chi-square tests of homogeneity and Mann–Whitney U tests were undertaken in SPSS Version 26, created by IBM corporation in Armonk, NY, USA, statistical suite [41], while bagging procedures and logistic regression analysis were undertaken in the STATA Version 17, created by Stata corporation in College Station, TX, USA, statistical analysis suite [42].

## 3. Results

### 3.1. Suicidal Ideation Prevalence in the ACI

The initial dataset comprised 13,896 participant responses. However, for the purpose of the current analysis, data were restricted to those who reported employment, reducing the number of participants to *n* = 11,607. Additionally, while attempts were made to control for missing data where possible, participants who did not report on the key variables of focus (suicidal ideation presence, traditional masculine norm adherence and psychosocial job adversity) were removed from the sample leaving *n* = 11,132. Participants were split into two groups, namely those employed in the ACI (*n* = 1721) and those classified as employed in Other Industries (*n* = 9411), with characteristics detailed below in Table 1. The prevalence of suicidal ideation in the ACI was 7.3% (95% CI 6.1–8.7%) compared to 7.4% (95% CI 6.8–7.9%) in Other Industries. This difference was not statistically significant.

### 3.2. Differences in Psychosocial Job Adversity—ACI and Other Industries

Comparison was made, in the form of a chi square test of homogeneity, to assess if there were proportional differences in prevalence of psychosocial job adversity for those employed in the ACI (*n* = 1721) and those classified as employed in Other Industries (*n* = 9411). The prevalence of psychosocial job adversity in the ACI was 63.6% (95% CI 6.1–8.7%) compared to 57.2% (95% CI 6.8–7.9%) in Other Industries and a statistically significant difference in proportions of 0.06, *p* ≤ 0.001.

### 3.3. Differences in Total Masculinity Scale Endorsement—ACI and Other Industries

The total CMNI masculine norm scores were assessed as highly skewed, and therefore a Mann–Whitney U test was undertaken to determine if there were differences in endorsement of traditional masculine norms between those employed in the ACI and other industries. The median score for ACI was 28 compared to 27 for Other Industries, with the difference being statistically significant (*p* ≤ 0.001).

### 3.4. Factors Associated with Suicidal Ideation in the ACI

Only those reporting employment in the ACI were included in this section of analysis. The strength of associations between the main variables of interest and Suicidal Ideation are detailed below (Table 2). For the multivariable analysis, the process of variable selection, as well as the statistical method are detailed below in Figure 2. All variables were assessed for missing data and the variable alcohol use was removed from analysis due to high levels of missing data (8%) [43]. Linearity of the continuous variables with respect to the logit of the dependent variable was assessed via the Box-Tidwell procedure with a Bonferroni correction applied using all 25 terms in the model (12 continuous variables, 12 interaction terms and the constant) resulting in statistical significance being accepted when *p* < 0.002. All but one of the continuous independent variables (emotional control domain of masculinity) were found to be linearly related to the logit of the dependent variable. An inverse square root transformation was applied to this variable and following transformation a linear relationship with the dependent variable was found.

Table 3 details the final logistic regression model. Presence of psychosocial job adversity, higher scores on the CMNI domains of self-reliance and risk-taking, not being in a relationship and experiencing a significant life event were all associated with increased likelihood of suicidal ideation presence. Self-reported depression in the past 12 months was associated with a greater than six times increased odds of the likelihood of experiencing suicidal ideation. Presence of outliers were assessed (none identified), the model passed the Stata Linktest and a Hosmer-Lemeshow goodness-of-fit test (*X*^2^(8, *n* = 1702) = 7.87, *p* = 0.45) showed the good calibration of the model.

## 4. Discussion

### 4.1. Prevalence of Suicidal Ideation Presence in the ACI

The current study suggests no differences in prevalence of suicidal ideation for those employed in ACI in comparison to males employed in Other Industries. This is an interesting finding given the regularly reported increased suicide vulnerability for those employed in the ACI [3,4,5,6]. Findings are suggestive of potential differences in suicidal ideation subtypes (passive or active) or in later suicidal trajectories for ACI employees and highlights the need for future research to understand what occurs in this period to drive the disparity in suicide outcomes so often reported for the ACI (e.g., contribution of impulsivity or access to lethal means in later suicidal trajectory processes). Despite prevalence similarities, the authors suggest it remains a significant issue and one that may be leveraged effectively in prevention strategies and programs if understood more deeply. Further investigation, potentially using alternative comparison populations or qualitative methods that focus on the nature and experience of suicidal ideation for those employed in the ACI, as well as how best to support someone experiencing these thoughts, would be useful.

### 4.2. Prevalence of Psychosocial Job Adversity in the ACI

The results of the current study suggest an increased prevalence of psychosocial job adversities in ACI compared to the general employed male population. While the difference is small it does suggest that the ACI may have some significant workplace issues that require focus and further research so as to mitigate the negative outcomes, including suicidal behaviours, that are being purported as associated with the presence of these factors [3,4,5]. Further research is needed in this space to support this finding, potentially using other comparison groups, as well as more nuanced investigation, such as subgroup analysis, to create richer understandings of whether there are distinct areas withing the industry more susceptible to experience of psychosocial job adversity. Again, qualitative methos may also prove useful to understand the nature and experience of psychosocial job adversities for those employed in the ACI, as well as provide information on how best to mitigate these issues. Despite this the current findings highlight the potential for an increased level of psychosocial job adversity prevalence within the ACI and the need for both prevention groups and the industry at large to place focus on these and mitigate where possible to reduce the negative outcomes regularly associated with their presence.

### 4.3. Adherence to Traditional Masculine Norms in the ACI

The study findings do suggest those employed in the ACI adhere to traditional masculine norms more so than those of the general employed male population. At face value, this supports previous assertions of a culture within the ACI that encourages adherence to traditional masculine norms above and beyond wider social expectations [3,4,5]. However, while a statistically significant difference is reported, it can be suggested as fairly trivial, likely driven by the large sample size, rather than large variances indicative of cultural differences. Therefore, we suggest treading carefully when discussing the previously held assertion that the ACI embodies a culture that encourages increased adherence to traditional masculine norms, above and beyond what is currently experienced in the broader male population. Further research is needed in this space to support this finding, as previous research suggests problematic culture not only for suicidal trajectories but in other contexts such as high levels of apprentice bullying [44]. Future research may consider comparison across more distinct populations (e.g., construction compared to non-male dominated industries) or across sub-groups within the ACI (e.g., apprentices compared to tradesmen, younger employees compared to older) to create richer understandings.

### 4.4. Psychosocial Job Adversity, Adherence to Traditional Masculine Norms and Suicidal Ideation in the ACI

Initial analysis demonstrates the association between presence of psychosocial job adversity and increased adherence to traditional masculine norms, with suicidal ideation presence for those employed in the ACI. While inclusion of other relevant variables, along with the specific domains traditional masculine norms, reduced association strength, the current study highlights the role of psychosocial job adversity leading to, or resulting from, suicidal ideation presence for those employed in the ACI. While due to the cross-sectional nature of the research we cannot comment on causality, pathways through which experience of psychosocial job adversity may lead to suicidal ideation presence are theoretically understandable. Perceptions of not being able to effectively undertake job requirements or whether a role will exist in the future are all highly stressful experiences. With theoretical models of suicide suggesting responses to stressful situations play an important role in suicidal ideation development and trajectories it is understandable that psychosocial job adversity may lead to suicidal ideation [45,46]. Additionally, presence of psychosocial job adversity may elicit experience of other psychological constructs suggested as important in suicidal experiences such as thwarted belonginess (i.e., a state were ones need for social connectedness is unmet) or perceptions of burdensomeness (i.e., a mental state characterized by apperceptions that others would be better off without them [45,46]. While further research is required to create deeper understandings of the causal nature of the association, with longitudinal research a recommended approach, the current findings provide important information to both preventative programs and the wider ACI on the need to be aware of the problems associated with the presence of psychosocial job adversity. For example, it is recommended industry engages in reducing psychosocial job adversities where possible or at minimum mitigate their impact through introduction of wellbeing approaches to support employees such as increasing engagement with industry specific preventative and support groups.

While initial analysis demonstrated an association between the total scale score of adherence traditional masculine norm and suicidal ideation presence in the ACI, further analysis demonstrates the bulk of this association is driven by only two domains of this construct, being self-reliance and risk-taking. This finding is vital given the broad stroke statements regularly implicating adherence to traditional masculine norms and suicidal behaviours in ACI employees. While further research is needed to determine causal pathways, as well as the potential for domains of traditional masculine norms to be associated with later stages of suicidal trajectories, it highlights the likely need for nuanced discussions regarding traditional masculine norms association with suicidal ideation for ACI employees, rather than implicating the construct as a whole.

Again, due to the cross-sectional nature of this research, causality cannot be determined, however, potential pathways that may drive associations between self-reliance, risk-taking and suicidal ideation presence are theoretically understandable. In the case of self-reliance, to remain stoic or not seek assistance in the face of adversity, may create a situation where development of suicidal ideation can occur. Furthermore, as Pirkis et al. [19] eloquently suggests, self-reliance fits within current psychological theories of the suicidal process, hypothesising that a man who is normally self-reliant might experience heightened perceptions of defeat or humiliation, suggested as important psychological constructs in suicidal models, if this state is challenged or disrupted [45]. Similarly, other psychological states suggested as central in suicidal ideation development and behaviours, such as perceptions of entrapment, perceived burdensomeness or thwarted belonginess, may be increased if regular processes to deal with issues, such as solving problems without support from others, are challenged [19,45,46]. Despite causal determination limitations that may be addressed by longitudinal study designs, findings support the need for those aiming to prevent and educate the ACI regarding suicidal states to reinforce the potential problematic nature of increased adherence to self-reliance attitudes, behaviours, or meanings. Additionally, this finding supports the need for increased awareness, stigma reduction and normalisation regarding suicidal states to encourage help-seeking behaviours to mitigate the potential role of self-reliance. The association between risk-taking and suicidal ideation presence in the ACI is considered a unique finding when reviewing similar investigations across broader Australian populations [19,23]. However, despite other research not reporting this association, it should not be considered surprising given the long standing and regular association reported between risk-taking and suicidal behaviours [47,48,49]. While again causality cannot be determined due to study design, there are theoretical pathways that can be suggested. Theoretical models of suicide suggest the role of impulsivity in suicidal trajectories and research has highlighted the interrelated nature between risk taking and impulsivity [45,46,47,48,49,50,51]. It may be that those who have increased adherence to risk-taking norms are also more likely to be impulsive, both with their decision making and cognitive processes, leading to increased suicidal behaviours or thoughts. Additionally, models have also highlighted acquired capability to suicide as an important aspect in suicidal trajectories [45,46]. With one more likely to engage in risk-taking behaviour, factors suggested as relevant in one capability to suicide, including fearlessness of death and pain desensitisation, may be altered, subsequently increasing one’s capability to die [45,46]. However, despite the theoretical relevance of these casual pathway suggestions, these implicate risk-taking in later processes of the suicidal trajectory as opposed to suicidal ideation development. Unfortunately, minimal research has been undertaken to understand if there is in fact a causal pathway that leads one from increased adherence to risk-taking to suicidal ideation presence [51]. This is an area that needs further research and again lends itself to longitudinal study designs to understand the causal pathways of the association. Despite this need for further research, the study’s findings do inform those aiming to prevent and educate the ACI regarding suicidal states to reinforce the complications of adhering to risk-taking attitudes, behaviours, or meanings at all stages of the suicidal trajectory through education and awareness training.

### 4.5. Other Factors Associated with Suicidal Ideation in the ACI

Relationship status, experience of a serious life event and self-reporting of depression in the last twelve months were all associated with increased likelihood of suicidal ideation presence for ACI employees. While these findings are unsurprising, given they have been regularly suggested and empirically supported as known associates of suicidal ideation presence in wider populations, they do add support to their relevance in suicidal states for those employed in the ACI. Additionally, while guiding support and preventative programs to be aware of the associations and educate the industry as such, the findings also inform future research of the need to account for these variables in analysis so as to create more conclusive understandings of potential associations or drivers.

### 4.6. Limitations

There are limitations within this study that require acknowledgement. The study relied on cross sectional data and as highlighted causal pathways cannot be determined outside of theoretical suggestion. While future research of a longitudinal method would allow for increased knowledge, this was not possible for the current research, with the data set we had access to. While there are additional waves of assessment available for the database used in the current study, the measure of traditional masculine norm adherence was not used beyond the initial wave. Given the potential for variabilities in adherence to traditional masculine norms as a result of biopsychosocial changes such as age, it was considered inappropriate to consider the adherence to traditional masculine norms as a trait-based variable, therefore limiting ability to utilise longitudinal methods [52]. Other limitations include a reliance on self-report data, as well as a sampling strategy that potentially limited appropriate representations of rural/remote males, as well as those of culturally and linguistically diverse background. In addition, the use of the single PHQ9 item to capture current suicidal ideation does not delineate between “passive” and “active” suicidal ideation. As suggested in previous research this is defensible given the item is regularly used to screen for suicidal ideation with good specificity and reasonable sensitivity when compared to a structured clinical interview [19]. However, it’s use does limit understandings of potential differences in suicidal ideation prevalence when comparing the male ACI employees to other employed men. It may be that a reason for the increased suicide vulnerability seen in the ACI is in part the result of greater prevalence of “active” suicidal ideation, however the current study cannot determine this.

## 5. Conclusions

To the best of the authors’ knowledge, this is the first study to investigate prevalence of suicidal ideation in the ACI, prevalence of psychosocial job adversities, differences in adherence to traditional masculine norms, as well as the associations between the industry factors of psychosocial job adversity, adherence to traditional masculine norms and suicidal ideation presence for ACI employees. Findings suggest no differences in suicidal ideation prevalence for the ACI in comparison to other employed Australian men. These findings suggest a need for more research to understand reasons for increased ACI suicide vulnerability, with potential for unique challenges, related to later stages of suicidal trajectories, to be driving this disparity. The current study demonstrates the increased prevalence of psychosocial job adversities for ACI employees in comparison to other employed Australian men. While further research is needed, it does highlight the need for prevention group and industry focus on these issues to mitigate negative outcome associated with their presence including that of suicidal ideation and behaviours. While statistically significant differences between ACI and other employed males regarding adherence to traditional masculine norms is reported, they are trivial and likely not indicative of cultural differences. Subsequently, the previous assertion that the ACI embodies a culture that encourages increased adherence to traditional masculine norms above and beyond what is currently experienced by the broader male population, may need reformulating and at minimum requires further investigation. It would be useful to compare the prevalence of these norms and suicidal ideation in the construction industry, with males from other industries such as agriculture, who are also thought to adhere to traditional masculine norms and be at heightened risk of suicide [53,54]. The current research demonstrates associations between psychosocial job adversity as well as the self-reliance and risk-taking domains of traditional masculine norms and suicidal ideation presence for ACI employees. While casual determination is not possible, it does highlight to both prevention groups and the wider ACI the need to be aware of these factors when discussing suicidal ideation and the role they play in its presence, as well as the importance of addressing these factors were possible, both through industry changes to reduce presence of psychosocial job adversities, as well as through stigma reduction and education activities regarding suicidal states. Finally, while further research is needed, findings indicating only self-reliance and risk-taking domains of traditional masculine norms are associated with suicidal ideation presence in the ACI, highlights the need for more nuanced discussion regarding adherence to traditional masculine norm’s role in suicidal ideation and possibly later trajectories, as opposed to previous indictments of the construct as a whole.

## Figures and Tables

**Figure 1 ijerph-19-15760-f001:**
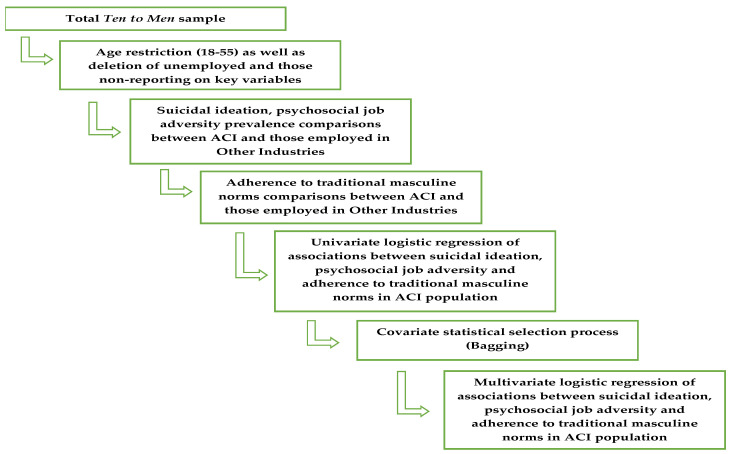
Methodology Flowchart.

**Figure 2 ijerph-19-15760-f002:**
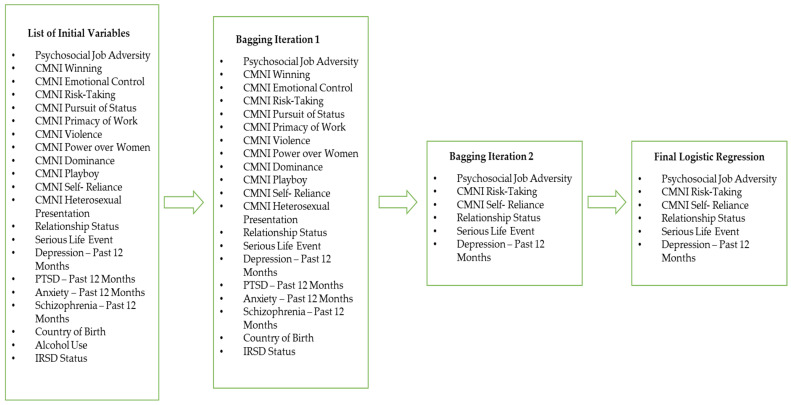
Logistic Regression Variable Selection Procedure.

**Table 1 ijerph-19-15760-t001:** Key Variable Characteristics of ACI and Other Industries.

	Australian Construction Industry Workers			Other Industries		
Characteristic	*n* (%)	Range	Mean (SD)	*n* (%)	Range	Mean (SD)
Suicidal Ideation	1721			9411		
Present	126 (7.3)	-	-	694 (7.4)	-	-
Not Present	1595 (92.7)	-	-	8717 (92.6)	-	-
Psychosocial Job Quality	1721			9411		
Psychosocial Job Adversities	1095 (63.6)	-	-	5386 (57.2)	-	-
No Psychosocial Job Adversities	626 (36.4)	-	-	4025 (42.8)	-	-
**Traditional Masculinity Domains**						
Winning	1721	0–6	2.49 (1.07)	9411	0–6	2.45 (1.06)
Emotional Control	1721	0–6	3.30 (1.27)	9411	0–6	3.15 (1.35)
Risk-Taking	1721	0–6	2.85 (1.17)	9411	0–6	2.73 (1.16)
Pursuit of Status	1721	0–6	3.24 (1.01)	9411	0–6	3.33 (1.02)
Primacy of Work	1721	0–6	2.58 (1.13)	9411	0–6	2.58 (1.19)
Violence	1721	0–6	2.49 (1.44)	9411	0–6	2.34 (1.46)
Power over Women	1721	0–6	1.41 (1.00)	9411	0–6	1.23 (1.00)
Dominance	1721	0–6	2.55 (1.09)	9411	0–6	2.47 (1.08)
Playboy	1721	0–6	1.62 (1.34)	9411	0–6	1.54 (1.32)
Self- Reliance	1721	0–6	2.69 (1.13)	9411	0–6	2.52 (1.10)
Heterosexual Presentation	1721	0–6	3.12 (1.54)	9411	0–6	2.85 (1.57)
Total Scores	1721	5–56	28.33 (5.58)	9411	7–61	27.20 (5.49)
**Age**	1721	18–57	37.95 (10.27)	9408	18–55	38.93 (10.14)
**Relationship Status**	1712			9320		
Married/Defacto	1217 (71.1)	-	-	6786 (72.8)	-	-
Unpartnered/Divorced/Widowed	495 (28.9)	-	-	2534 (27.2)	-	-
**Depression–Past 12 Months**	1693			9259		
Present	170 (10.0)	-	-	990 (10.7)	-	-
Not Present	1523 (90.0)	-	-	8269 (89.3)	-	-
**PTSD–Past 12 Months**	1677			9237		
Present	11 (0.7)	-	-	131 (1.4)	-	-
Not Present	1677 (99.3)	-	-	9106 (98.6)	-	-
**Anxiety–Past 12 Months**	1688			9240		
Present	93 (5.5)	-	-	603 (6.5)	-	-
Not Present	1595 (94.5)	-	-	8637 (93.5)	-	-
**Schizophrenia–Past 12 Months**	1692			9232		
Present	4 (0.2)	-	-	20 (0.2)	-	-
Not Present	1688 (98.8)	-	-	9212 (98.8)	-	-
**Significant Life Event**	1711			9333		
Present	663 (38.7)	-	-	3335 (35.7)	-	-
Not Present	1048 (61.3)	-	-	5998 (64.3)	-	-
**Country of Birth**	1719			9278		
Australia	1416 (82.4)	-	-	7108 (75.8)	-	-
Other	303 (17.6)	-	-	2270 (24.2)	-	-
**Alcohol Use**	1599			8719		
Appropriate	779 (50.0)	-	-	2952 (33.9)	-	-
Inappropriate	780 (50.0)	-	-	5767 (66.1)	-	-
**IRSD**	1721			9410		
Level 1 (Greatest Disadvantage)	248 (14.4)	-	-	1481 (15.7)	-	-
Level 2	309 (18.0)	-	-	1699 (17.7)	-	-
Level 3	417 (24.2)	-	-	2296 (24.4)	-	-
Level 4	373 (21.7)	-	-	2128 (22.6)	-	-
Level 5 (Least Disadvantage)	374 (21.7)	-	-	1836 (19.5)	-	-

**Table 2 ijerph-19-15760-t002:** Univariate Logistic Regression Assessing Associations between Suicidal Ideation and the Domains of Masculinity and Psychosocial Job Adversity in the ACI.

Variable	Odds Ratio	Standard Error	95% CI		*p*
			Upper	Lower	
Psychosocial Job Adversity	2.06	0.46	1.33	3.20	<0.001
Traditional Masculinity Overall	1.04	0.17	1.01	1.08	<0.001

**Table 3 ijerph-19-15760-t003:** Multivariate Logistic Regression Assessing Associations between Suicidal Ideation, Masculinity Domains of Self-Reliance, Risk Taking and Psychosocial Job Adversity.

Variable	Odds Ratio	Standard Error	95% Confidence Interval		*p*
			Upper	Lower	
Psychosocial Job Adversity	1.79	0.42	1.12	2.85	0.01
Risk Taking	1.20	0.10	1.01	1.41	0.03
Self-Reliance	1.29	0.11	1.09	1.51	0.00
Relationship Status	1.54	0.32	1.03	2.32	0.04
Serious Life Event	1.62	0.34	1.07	2.43	0.02
Depression–Past 12 months	6.88	1.49	4.50	10.51	0.00

## Data Availability

The data used in this study is available via submission to the Australian Institute of Family Studies with further information available at https://tentomen.org.au/, accessed on 28 October 2022.

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
