# Peer review of "Suicidal Ideation in the Australian Construction Industry: Prevalence and the Associations of Psychosocial Job Adversity and Adherence to Traditional Masculine Norms"

_ijerph, 2022, doi:10.3390/ijerph192315760_

Round 1
Reviewer 1 Report
Thank you for your submission. I understand this is the first study to investigate prevalence of suicidal ideation in the ACI, prevalence of psychosocial job adversities, differences in adherence to traditional masculine norms, as well as the associations between the industry factors of psychosocial job adversity, adherence to traditional masculine norms and suicidal ideation presence for ACI employees.
Through my reviewing, I would like to give several comments to be revised.
[Abstract]
- In this study results, it would have been better to show the statistical values (e.g. p-value, odds rate) to understand easily. Besides, it is better to write using structural writing such as "Background", "Aim", "Methods", "Result" and "Conclusion."
[Introduction]
- Why "it is unsurprising that those employed in the male-dominated Australian Construction Industry (ACI) are regularly reported to have increased suicide vulnerability"? Could you please show additional explanations in this matter?
- It would be better to clarify more why the author(s) had to focus the study subjects on ACI (e.g., ACI workers have a higher suicide rate than workers in other industries).
- Research hypothesis: In this study, what is the research hypothesis? To understand easily for readers, it should mention the research hypothesis in the introduction section. (e.g., High adherence to traditional masculine norms is related to the high prevalence of suicidal ideation among Australian construction industry workers)
[Methods]
- It is necessary to show the detailed information of "Ten to Men", not just show the URL. Visiting the website of "Ten to Men", I was able to understand at last that Ten to Men is the Australian Longitudinal Study on male health is a study designed to help improve the health and well-being of boys and men.
Also, in 2013–2014 (Wave 1 of the study), Ten to Men collected health and lifestyle information from nearly 16,000 boys and men across the country via surveys and interviews. Study participants were randomly chosen to ensure a broad range of backgrounds and life experiences.
- Regarding measurements of "Psychosocial Job Adversity" and "Adherence to Traditional Masculine Norms", to ensure the reproducibility of this research, it would be better to show the actual questions in the appendix.
- Besides, is "Adherence to Traditional Masculine Norms" a validated scale?
[Results]
- What is the breakdown of other industries? I think that many industries are included in the "other", therefore it seems that it is necessary information when considering the background of the findings.
- By the way, did the author(s) conduct analysis multivariate logistic regression analysis between suicidal Ideation and the domains of masculinity and psychosocial job adversity in the other industries? I think the comparison of the result in multivariate logistic regression analysis between the ACI and the other industries would be required.
[Discussion]
- I think that setting a control group (other industries) is vague, therefore it would be caused no relationship of suicidal ideation between ACI and others. The more clear study design is required.
- Based on your findings that strong psychosocial job adversity and adherence to traditional masculine norms in ACI have a significant association with suicidal ideation, I would appreciate it if the author(s) could mention a little more deeply and could make specific suggestions for overcoming these challenges.
Author Response
We thank the reviewer for the comments and feedback. Please see that attachment for our responses.

Reviewer 2 Report
Review of the paper: Suicidal Ideation in the Australian Construction Industry: Prevalence and the Associations of Psychosocial Job Adversity and Adherence to Traditional Masculine Norms
The article is well and logically written. The paper feature a separate subchapter devoted to the description of the methodology. The results presented and commented in sufficient details with sound reasoning and appropriate interpretation.
Despite the undoubted merits of the article, I also have comments that are worth answering:
- I suggest you be clearer in your abstract about your research implications/limitations, and practical contribution;
- It is suggested to present the structure of the article at the end of the introduction;
- A flowchart might be added to the article to show the research methodology;
- Methodology should include dates of the research;
- The question of whether other authors have done systematic review in this area of research is worth answering.
Author Response
We thank the reviewer for their feedback and comments. Please see attached for our reponses.

Round 2
Reviewer 1 Report
I have confirmed that the author(s) have revised it appropriately.